# Preterm Birth and Its Association with Maternal Diet, and Placental and Neonatal Telomere Length

**DOI:** 10.3390/nu15234975

**Published:** 2023-11-30

**Authors:** Nikoletta Lis, Demetris Lamnisos, Aikaterini Bograkou-Tzanetakou, Elena Hadjimbei, Irene P. Tzanetakou

**Affiliations:** 1Department of Health Sciences, European University Cyprus, Nicosia 2404, Cyprus; nikolettalis@yahoo.gr (N.L.); d.lamnisos@euc.ac.cy (D.L.); 2Maternity Clinic, Cork University Maternity Hospital, T12 YE02 Cork, Ireland; 3School of Medicine, European University Cyprus, Nicosia 2404, Cyprus; k.tzanetakou@external.euc.ac.cy; 4Department of Life Sciences, European University Cyprus, Nicosia 2404, Cyprus; e.hadjimbei@euc.ac.cy

**Keywords:** maternal nutrition, placenta telomeres, newborn telomeres, premature birth, preterm infants

## Abstract

Preterm birth (PTB), a multi-causal syndrome, is one of the global epidemics. Maternal nutrition, but also neonatal and placental telomere length (TL), are among the factors affecting PTB risk. However, the exact relationship between these factors and the PTB outcome, remains obscure. The aim of this review was to investigate the association between PTB, maternal nutrition, and placental-infant TL. Observational studies were sought with the keywords: maternal nutrition, placental TL, newborn, TL, and PTB. No studies were found that included all of the keywords simultaneously, and thus, the keywords were searched in dyads, to reach assumptive conclusions. The findings show that maternal nutrition affects PTB risk, through its influence on maternal TL. On the other hand, maternal TL independently affects PTB risk, and at the same time PTB is a major determinant of offspring TL regulation. The strength of the associations, and the extent of the influence from covariates, remains to be elucidated in future research. Furthermore, the question of whether maternal TL is simply a biomarker of maternal nutritional status and PTB risk, or a causative factor of PTB, to date, remains to be answered.

## 1. Introduction

Preterm birth (PTB), is defined as the birth whose onset occurs before the 37th week of pregnancy [1]. It commences automatically with the presence of one or more of the following events: uterine contractions without rupture of membranes, premature rupture of membranes, induction of labor or the need of a caesarean section. Induction of preterm labor occurs in the presence of pregnancy-related pathology, such as preeclampsia and intrauterine growth restriction (IUGR), where the fetuses’ delivery is necessary and/or urgent [2].

PTB is truly a global problem. An estimated 13.4 million infants were born prematurely in 2020, which translates to more than 1 in 10 babies, with the highest percentages reported in southern Asia and sub-Saharan Africa [1]. Adding to this gloomy statistic, most countries show an increased rate of PTBs over the last 20 years [3]. Prematurity is considered a multi-causal syndrome and one of the global epidemics [3]. Goldenberg et al. [2] identified prematurity as a syndrome, due to the contribution of multiple possible causes for its spontaneous onset, including infection or inflammation, vascular diseases, uterine hypertension (e.g., twin pregnancy, polyhydramnios), etc.

The multiple risk factors implicated in premature labor include lifestyle, heredity, anthropometric characteristics, multiple pregnancy, and maternal age [4]. Other main causes of PTB receiving increased scientific interest recently include placental premature cellular aging and/or a dysfunctional placenta [5]. Moreover, the cervixes’ short length in the second trimester of pregnancy (<2.50 cm) and the increased concentration of fetal fibronectin in the cervical smear, are possible prognostic factors of spontaneous onset of PTB [4,6].

Even to this day, with the technological and research advances recorded, the causes, and mechanisms of the spontaneous onset of both preterm and full-term birth, remain poorly understood [1,7]. It is thought, however, that the spontaneous onset of both term- and preterm labor share the same mechanism. The difference is that the onset of full term labor is the result of a physiological maturation process, while the premature one derives from a pathological process that activates one or more of the above stages of the mechanism [8].

The placenta, which develops throughout pregnancy, has a key role in a variety of functions including labor outcome. It acts as an endocrine gland, an immunomodulatory organ, it is part of the fetal circulatory system, while it also participates in the nutrient/metabolite and oxygen transportation to and from the fetus [9]. Its anatomical formation separates the fetal from the maternal circulation, thus highlighting the importance of “filtering” both macro- and micronutrient transportation [10], while the placenta’s fetomaternal nutrient transportation ability depends on its size, weight, and morphology [11,12,13,14]. However, the mechanism is, in essence, much more complex.

As the pregnancy progresses, the placenta undergoes aging-related changes. During this process, its cells display reduced functionality, mitochondrial dysfunction, and finally apoptosis. Placental aging is a normal phenomenon, which was first described in the 70s, but only during the latest decade has the association between placental aging and various changes during pregnancy been studied. Nevertheless, in some cases the placental cells’ aging occurs prematurely, disrupting their normal function [15,16]. Various factors may affect the pace of aging of the placenta, proper fetal development, and birth outcomes, including maternal dietary intake and nutritional status [17,18,19,20]. Not surprisingly, maternal nutritional status—among other things, through its effect on the placenta—is associated with the onset of spontaneous PTB, low birth weight and irregular fetal growth [11,21,22,23,24,25,26,27,28,29,30]. In a majority of studies, beneficial dietary patterns that are often characterized by high consumption of nutrient dense foods, including vegetables, fruits, whole grains, fish and dairy products, appear to have a synergistic effect on reducing the risk of PTB. Unhealthy dietary patterns, on the other hand—such as the ones characterized by high intakes of refined grains, processed meat, and foods high in saturated fat or sugar—are associated with a higher risk of PTB [11]. However, the exact mechanism/s through which the characteristics of the diet (e.g., quality, quantity, intake timing, and frequency) affect the risk of PTB, are a complex research matter still under investigation.

Another factor, linked to both placental aging pace and PTB, is telomere length (TL). Telomeres, used as potential indicators of the aging pace and disease burden, are repetitive, non-coding DNA sequences found at the ends of eukaryotic chromosomes that protect the integrity of DNA information throughout the cell cycle, preventing loss of DNA during cell division [31]. Telomeres are susceptible to oxidative stress damage, genetic/epigenetic, and environmental factors including unfavorable nutritional habits, leading to premature length reduction and the subsequent development of chronic diseases and/or premature death [32,33].

Pregnancy is on its own characterized as a state of high oxidative stress. Indeed, as the metabolic demands of the developing fetus increase, the production of reactive oxygen species (ROS) rises. However, a persistent, intense, and premature oxidative stress affects the placenta’s antioxidant capacity, followed by an accumulation of ROS and damage to lipids, proteins, and the placenta’s DNA. Inductively, the placental cells undergo aging and erosion of their TLs. Shorter TL has been associated with pregnancy complications, such as maternal and fetal mortality, preeclampsia, IUGR, and PTB [34]. Additionally, placental TLs of PTBs are reported to be shorter compared with the placentas of full-term pregnancies. This mechanism speculates that the TL is associated with the premature placenta’s aging pace, as well as the adverse outcome of premature birth [35].

In recent years, TL and telomerase (i.e., the enzyme responsible for maintenance of the length of telomeres by the addition of guanine-rich repetitive sequences), have been investigated as possible biomarkers of premature placental aging [15,16]. Physiologically, cell division is accompanied by TL shortening that leads towards the aging trajectory. On the other hand, the enzyme telomerase is active in ensuring the preservation of placental TL and subsequently its normal function until the end of pregnancy. This balanced process can be affected by various aggravating factors mentioned above (e.g., oxidative stress, inflammation, and other genetic/epigenetic, immunological, physiological, lifestyle, and environmental factors including nutrition), leading to an advanced aging pace and the consecutive dysfunction of the placenta [15,36]. As a result, complications during pregnancy, primarily gestational diabetes mellitus (GDM), preeclampsia, intrauterine growth restriction (IUGR), PTB, and intrauterine death, arise [35,37]. Indeed, the placentas of complicated pregnancies with preeclampsia, intrauterine growth retardation, are characterized by decreased activity of telomerase and generalized dysfunctionality of TL regulation. In complicated pregnancies, a short placental TL is recorded, but no similar findings are evident in the TLs of umbilical cords. On the contrary, molar pregnancies have shown an increased telomerase activity, similar to the telomerase activity recorded in malignant cells and tissues [38].

The effect of the above influences is depicted in the offspring, and although the majority of prior telomere research has focused on adults, TL attrition in early life may be particularly important for lifelong health status, as evident from studies of childhood adversities and their effect on adult health trajectory [39].The importance of early life TL programming is further evident if one considers that TL in adulthood is determined by both TL at birth (which serves as an “initial setting”) and by its subsequent attrition during development [40].

Since TL shortening is a biological phenomenon associated with many adverse health conditions, including various complications associated with pregnancy [41] and the subsequent health trajectory of offspring, it may serve as a reflection of the cumulative impact of stressors and/or as part of a fetal programming mechanism [42]. Indeed, newborn TL, seen as the primary setting of TL regulation, although highly variable among individuals [42], carries important lifetime consequences for telomere dynamics and longevity [43,44]. Recent evidence regarding newborn TL suggests that plasticity exists in both the programming of telomere biology as well as the initial setting of TL after birth [45]. The “in utero” environment appears to be a significant contributor to this effect. Indeed, cord blood TL association studies with specific fetal exposures to maternal nutrient status, e.g., folate [46] and vitamin D [47], smoking status [48], maternal educational profile [49], or maternal metabolic status [50,51,52], and parental age [52,53,54,55,56], have provided insights into the mechanism of TL regulation and its trajectory following birth. Maternal stress [57], stress hormones [58], adverse early life events [39], and other related factors have been correlated with TL and point to the importance of in utero and early developmental exposures in the regulation of offspring TL in adult life. As shown by Gotlib et al. [59], chronic stress is a major determinant of telomere maintenance, both by direct exposure and through pathways of mother-to-fetus/infant transmission in early life.

Previous studies have shown that premature infants exhibit longer telomeres at birth [60,61], but shorter telomeres in young adulthood compared with term-born infants [62,63], suggesting that a more rapid attrition occurs in early life [64,65]. From this perspective, telomere regulation might be a key programming and compensatory mechanism, in premature infants. Furthermore, postnatal exposure to adversities associated with preterm birth are a cause for enhanced TL attrition. In particular, preterm infants are characterized by an immature neurobehavioral profile at birth, even in the absence of severe brain injuries and associated perinatal complications. Therefore, “by definition” they require long-lasting hospitalization in the Neonatal Intensive Care Unit (NICU). This increases the newborns’ chances of survival but, at the same time, entails a number of stressors, such as the physical, pain related, and socioemotional, thus representing an early adverse experience, linked to detrimental consequences for neurological, neuro-endocrinal, sensorial, behavioral, emotional, and socioemotional development, as well as to increased chronic disease risk, later in life [60,66]. Indeed, elevated oxidative stress and inflammation, both of which contribute to telomere shortening, has been recorded in premature hospitalized preterm infants, while a measurable decline in TL during hospitalization, may convey information about NICU exposures that carry both immediate and long-term health risks [67,68,69,70]. It is therefore not surprising that researchers assume preterm infants display an “aged” phenotype [61], further increasing the importance of TL regulation and monitoring. Nevertheless, whether infant and placental TL are biomarkers of PTB, or the actual cause of this adverse outcome, remains to be discovered.

The aim of this literature review is to investigate the possible associations between maternal nutrition, placental/newborn TL and PTB.

## 2. Materials and Methods

### Description of Search Strategy

This is a review study of the international scientific literature. Articles published in scientific journals were sought from the online databases PubMed and EBSCO host, from 2013 to 2023, in an effort to reflect current knowledge and well-established research trends. The search took place from March to September 2023 and the terms used, were:(Maternal OR pregnancy OR gestational OR fetal);

AND 

(Nutrition OR diet OR nutrients);

AND 

(Placenta OR newborn OR infant OR neonatal OR offspring OR fetus);

AND 

(Telomeres OR telomere length));

AND 

(Preterm OR premature);

AND 

(Birth OR labor);

AND 

(Observational studies OR cohort studies OR case control studies OR cross-sectional studies OR case studies).

The searches were conducted independently by two reviewers, namely NL and IT. Initially, the abstracts were read. In this way, research selection was directly related to the subject. Citation chaining also took place in order to avoid omitting relevant studies. With the completion of the article searches, the studies were classified into thematic categories based on the exposure and the outcome (PTB, maternal nutrition, placental, and neonatal TL).

Study inclusion criteria:Studies published from 2013 to 2023;Observational studies (i.e., cohort, case–control, cross-sectional studies);Direct relation to the subject and inclusion of the specified keywords;Conceptual connection of keywords with the title and summary;Inclusion of studies with a positive, negative, or neutral effect on the association of maternal nutrition with placental/newborn TL and PTB;Description of the search strategy and eligibility of studies according to the four-phase flow chart of the PRISMA guidelines [71].

## 3. Results

The search returned several studies for each keyword separately. However, when all keywords were combined together, no studies were found associating maternal nutrition with placental and/or newborn TL and PTB.

For this reason, the keywords A: maternal nutrition, B: PTB, and C: placental/newborn TL, were searched in dyads as follows: concept “A” relation to concept “B”, “C” to “A”, and “C” to “B”. It is impossible, however, to find a chronological sequence of the exposure and the outcome, but evidence of associations between these factors would imply a potential relationship (either causal or correlational) [72].

### 3.1. Studies Eligibility

As previously mentioned, from the thorough preliminary search of the relevant literature in PubMed and EBSCO host electronic databases, no studies with the combination of keywords were found. The process of selecting and excluding the studies was carried out according to the PRISMA guidelines, as shown in Figure 1 [71].

Thus, the literature search was performed linking concepts A to B, B to C, and A to C. In total, 40 observational studies containing the key words in the combinations described above were included.

The selected studies were evaluated for their methodological quality on the basis of the checklist of the 22 thematic modules, examined by the STROBE scale, and were all considered to be of high quality.

### 3.2. Maternal Nutrition and PTB

Of the above 40 included studies, 21 were found to associate maternal nutrition with premature birth. The type of study, population characteristics and outcome, of each publication included in this section, may be seen in Table 1. Their publication years were from 2014 to 2023. Five of the studies took place in China, one in Indonesia, one in Norway, two in Singapore, three in the United States of America, one in Canada, one in Malawi, one in Pakistan, one in Mexico, one in Brazil, one in Switzerland, one in Portugal, one in Bangladesh, and one in Italy.

Regarding the methodology, 12 of the studies concerned cohort studies, four studies were cross-sectional and five studies were case–control. Overall, the studies’ populations were pregnant women with a singleton pregnancy and the absence of any pathology or pregnancy complications.

Dietary intake was assessed with validated questionnaires, and/or measurements of anthropometric characteristics of the pregnant woman (BMI, arm circumference, etc.), as well as the newborn (sex, weight, length). Also, delivery characteristics, such as gestational age (GA) at delivery, were documented.

The results showed both positive and negative associations between maternal nutrition (either as a dietary pattern of adherence or individual food items, or supplement/s intake) and PTB.

Adherence to a healthy diet, as recorded by a maternal nutritional score and GA, at birth, were positively correlated. In particular, a significant increase in the risk of PTB was associated with low nutritional scores from a questionnaire developed and based on the International Federation of Gynecology and Obstetrics (FIGO) recommendations on adolescent, preconception, and maternal nutrition. Furthermore, a significant increase in preterm deliveries, with a relative risk of 1.44, was recorded for women with a first trimester nutritional score lower than five. However, the single food items of the score calculation were not associated with either early placental markers or complex pregnancy outcomes [73].

The Norwegian Fit for Delivery diet score, either in pre-pregnancy or early pregnancy, was protectively associated with excessive GWG and risk of PTB. The protective association with high birthweight was confined to pre-pregnancy diet and with preeclampsia to early pregnancy diet. The study recorded no association between the pre-pregnancy diet score and preeclampsia [74].

Dietary patterns related to reduced food intake or fasting in the 2nd trimester of pregnancy (the Ramadan period of the Islamic religion), were positively associated with increased risk of very PTB between 28 and 31 weeks of gestation. At 15–21 weeks, the risk increased by 1.33 times and at 22–27 weeks by 1.53 times [75].

Yogurt consumption during pregnancy appeared to have a negative (protective) association with PTB, especially when combined with a normal BMI [76]. However, in the study by Lu et al. [77] higher dairy consumption compared to vegetable consumption, showed a strong correlation with PTB.

In very PTB, the maternal dietary consumption during pregnancy of Portuguese women giving birth very prematurely was assessed. Consumption of certain foods did not comply with the recommendations for pregnant women by the Portuguese Directorate General of Health. In particular, consumption was below the recommended levels for dairy products (one vs. three portions), vegetables (two vs. three portions), and fruits (one vs. four portions). The study also recorded a very low cereal intake (average of one portion per day ingested). Furthermore, PTB-associated pregnancy-induced hypertension was associated with increased consumption of pastry products, fast food, bread, pasta, rice, and potatoes. However, only bread consumption had a weak but statistically significant association with pregnancy-induced hypertension in a multivariate analysis model [78].

A fruit/vegetable/rice-based dietary pattern during pregnancy appeared protective against PTB and small for gestational age (SGA) infants, compared with two other dietary patterns; namely, one based on consumption of seafood and noodles and the second on consumption of processed meat, cheese, and pasta. Only the fruit/vegetable/rice pattern contributed positively to the development of long for gestational age (LGA) neonate [79].

In the prospective cohort of Martin et al. [80], diet quality during pregnancy was associated with PTB. Specifically, greater adherence to a healthy dietary pattern, such as the DASH diet, reduced the chance of PTB, especially during the second trimester. Conversely, a greater adherence to a dietary pattern of poorer quality, such as pre-processed or fast foods, high-fat foods, and confectionary, increased the chance of PTB.

The use of folic acid supplements of 400 μg or more showed a negative correlation with PTB. Folic acid use appeared to reduce PTB by 14%. Strong association with spontaneous PTB and non-significant for iatrogenic PTB or prolonged spontaneous rupture of membranes was also recorded [81]. Furthermore, the large cohort study of Wu et al. [82] showed similar results with folic acid periconceptional supplementation associated with a lower risk of PTB. In particular, the earlier women started taking their folic acid supplements prior to pregnancy (i.e., at least 3 months before their last menstrual period), the more likely to reduce the risk of PTB, compared with women who started taking folic acid later (i.e., 1–2 months before their last menstrual period).

Multivitamin use in the third trimester of pregnancy, had a negative correlation with preterm and very PTB, especially in pregnant women of African, non-Hispanic, descent. However, the dose–response effect was not investigated [83]. Similar results by Olapeju et al. [84] support that multivitamin supplement intake of at least three times/week throughout pregnancy is significantly associated with a reduction in the likelihood of PTB. Use during the third trimester was especially associated with a greater reduction in PTB risk than use in the first trimester, but there was no significant association between preconception supplement intake and PTB. Also, higher plasma folate levels were associated with lower risk of PTB.

In a case–control study, Ren et al. [85] examined hair levels of trace elements, including endocrine disrupting metal(loid)s (EDMs), such as lead, mercury, arsenic, and cadmium, and nutritional trace metal(loid)s (NTMs) such as zinc, iron, copper, and selenium, in 415 control pregnant women with birth at term and 82 pregnant women with PTB. Negative (protective) correlation between increased levels of NTMs, especially Fe and Zn and PTB occurrence, was recorded. Also, the potentially protective effect of mercury was seen for PTB, while for EDMs, only maternal hair mercury was negatively associated with PTB risk.

PTB was associated with a higher serum concentration of heavy metals (such as mercury and lead and lower maternal serum concentration of AtRA) and all micronutrients, and with lower placental concentration of manganese, iron, copper, zinc, selenium, AtRA, 25(OH)D, and higher placental concentration of mercury and lead. Compared with the PTB group, the term birth group had higher concentrations of copper and AtRA in cord blood [86]. Copper in maternal serum concentrations has been recorded above the upper normal limit in both term and PTB groups of a nested case–control study in Malawi. At the same time, PTB was associated with higher maternal serum concentrations of copper and zinc [87].

In the cohort study of Perveen and Soomro [88] iron deficiency anemia (Hb < 11 mg/dL) appeared positively associated with PTB, low birth weight, fetal mortality, and low Apgar score in the 1st and 5th minutes of birth. Additionally, increased levels of folic acid, in the third trimester were associated with reduced risk of PTB and longer duration of gestation of PTB. Little or no correlation was found between increased levels of B6 and B12 and PTB or SGA [89].

In a cross-sectional study by Christoph et al. [90], the lower the maternal serum vitamin D level, the higher the GA at birth, but no association was observed between vitamin D levels and PTB. In addition, in a nested case–control study in Bangladesh, vitamin D deficiency, common in Bangladeshi pregnant women, was associated with an increased risk of PTB [91].

An inverse relationship between maternal total protein levels (via diet) and PTB during the third trimester of pregnancy, especially in female fetuses, was recorded in a large Chinese cohort study [92]. The likelihood of adverse outcomes was higher in non-white obese women with high protein consumption, in the nested case–control study by Miele et al. [93] The anthropometric classification of obesity had a greater impact on PE and GDM, in contrast to PTB and SGA. In total, obesity had a small effect on PTB.

### 3.3. Maternal Nutrition and Placental and Newborn Telomeres

In total, 13 studies were retrieved and included in this review investigating the relationship between maternal nutrition and placental and newborn telomeres. The studies type, population characteristics and outcome may be seen in Table 2.

The studies’ publication years were from 2015 to 2022. The studies were conducted in various geographic locations, namely China (*n* = 1), Argentina (*n* = 2), Italy (*n* = 1), USA (*n* = 4), Belgium (*n* = 1), Seoul (*n* = 1), Seychelles (*n* = 1), Rwanda (*n* = 1), and Singapore (*n* = 1).

In terms of methodology, 12 were cohort studies and one was cross sectional. The study populations primarily consisted of pregnant women, while others were mother–newborn dyads. The basic characteristics of the majority of the studies’ populations concerned pregnant women in the first trimester of pregnancy with singleton pregnancy and the absence of pathology.

Blood samples of pregnant women, questionnaires monitoring their eating habits, supplement intake and/or measurements of anthropometric characteristics of the pregnant woman (BMI, arm circumference), of the newborn (gender, weight, length), and delivery characteristics (GA at delivery), were recorded. In addition, observations were made with regard to the characteristics of the placentas, measurements of their TLs as well as the leucocyte TL (LTL) of the newborns from samples of umbilical cord blood at delivery.

The results showed both positive and negative associations between maternal diet and placental and neonatal TL.

In terms of dietary lipids, one study examined the effect of n3 intake on umbilical cord, placenta, and infant TL. Maintaining higher levels of maternal n3 polyunsaturated fatty acids (PUFAs) during pregnancy may help maintain TL in the offspring, which is beneficial to long-term offspring health [94].

No clear associations were recorded for prenatal or postnatal PUFAs status and methylmercury exposure with offspring TL. A higher prenatal n6:n3 PUFA ratio was, however, associated with longer maternal TL [95]. On the contrary, a maternal high-fat dietary consumption pattern during pregnancy was associated with shortened TL among fetuses, after accounting for the effects of potential covariates [96].

One study correlated placental and umbilical cord blood TL with maternal nutritional profile. Specifically, a positive association between plasma vitamin 25(OH)D3 and placental TL was observed, while an inverse association was observed between BMI, body fat percentage, vitamin B12, and placental TL. Furthermore, a negative correlation was found between the above and the length of umbilical cord blood telomeres [97].

Furthermore, vitamin D levels during pregnancy, especially in the first trimester, were positively correlated with neonatal leukocyte TL at delivery [47,98]. A positive correlation was also seen between neonatal leukocyte TL and maternal leukocyte TL, maternal vitamin D concentration, maternal energy intake, and newborn weight [47].

A positive correlation was also seen between folic acid and neonatal TL. Each 10 ng/dL increase in pregnant women’s folic acid intake was associated with a 5.8% increase in mean TL. The average TL of newborns of mothers with low folate serum concentrations was shorter compared to those whose mothers belonged to the group with increased levels [46].

Furthermore, maternal folic acid supplementation after the first trimester and throughout pregnancy was associated with longer newborn TL [99]. However, in the same cohort, no significant association was found between maternal folic acid supplementation in the first trimester and newborn TL. One more study of the effect of folate on newborn TL showed a positive association between umbilical cord RBC folate and fetal TL at birth [100].

TL, but only in female rather than male newborns, was more susceptible to variation from maternal vitamin B12 levels, as well as maternal TL, and mental health. In total, maternal TL was strongly associated with antenatal factors, especially metabolic health, and nutrient status [65].

The increased exposure of pregnant women to toxic metals (antimony, lithium, arsenic) was positively correlated to placental TL and newborn sex. Antioxidants (zinc, selenium, folate, Vit D3), did not contribute to the modification of the above process. The TL of the placenta, decreases as the age of the pregnant woman increases, approximately by 1% per year. Lithium appears to increase the mother’s TL. Lead (in umbilical cord blood) showed an inversely proportional correlation with infant TL, particularly in male neonates [101].

Magnesium deficiency was negatively associated with maternal RTL after adjusting for covariates. A positive association between maternal intake of magnesium and TL of cfDNA from amniotic fluid, while results on other micronutrients (i.e., vitamin B1 and iron) were marginally significant [102]. Also, Myers et al. [103] examined the relationship of vitamin C intake and cord blood TL. The sample similarly to the Louis-Jacques et al. [100] study was drawn as a part of an ongoing prospective cohort study conducted by the University of South Florida (USF), Morsani College of Medicine. A positive association between maternal vitamin C intake and fetal TL was observed.

### 3.4. PTB, Placental, and Newborn Telomeres

As for the association of PTB with both the placental and the newborn telomeres, six studies, published from 2015 to 2023, were found. The studies type, population characteristics and outcome may be seen in Table 3. The study countries were Brazil (*n* = 1), United Kingdom (*n* = 1), Indonesia (*n* = 1), Israel (*n* = 1), India (*n* = 1) and France (*n* = 1). In terms of methodology, two were cohort, one case–control, and three were cross sectional studies.

The population of the studies was heterogenous and included pregnant women in normal labor, mother–infant dyads, or 2- and 7–9-year-old children, who were born either full term or prematurely. Data analysis included anthropometric measurements, dietary intake questionnaires, observations regarding pregnancy course, DNA methylation of umbilical cord blood and placental tissue, umbilical cord blood, and placental tissue LTL.

A study indicating no effect of PTB on TL showed similar placental TL in PTB and term labor placentas. In this cross-sectional study, early telomere shortening in PTB, was observed, that mimics the term placenta. Markers 8-OHdG and HMGB1, did not correlate with placental telomere ratio, while HMGB1 from the placenta of both PTB and term labor showed no significant difference. The equal relative amount of telomeric DNA (T) to the beta-globin single copy gene (S), calibrated to a plate reference genomic DNA sample (T/S ratio) of the placenta from PTB and term labor was recorded [104].

In contrast, telomere shortening in fetal membranes, was suggestive of senescence associated with triggering of labor at term. As such, fetal membranes from the term labor group showed TL reduction compared with those from the others. However, telomerase activity did not change in fetal membranes, irrespective of pregnancy outcome [105].

In the case–control study by Vasu et al. [61] preterm infants showed longer TL than full-term infants. A positive correlation was observed between maternal age and the telomere/shelterin (protein) ratio, while the higher the age of the mother, the longer were the telomeres of the newborn (*p* = 0.011). Accordingly, maternal blood and placental samples from spontaneous PTB presented shorter telomeres and increased Gal-3 expression compared with the spontaneous term pregnancies group [106].

Another longitudinal cohort study tracked premature infants into adulthood by studying TL in saliva, as well as lung function. A positive correlation was recorded between TL and abnormal lung airflow in the adult population who were born prematurely. There was no apparent association with perinatal causes of PTB. Nevertheless, no apparent association with perinatal events and TL was noted [70].

Increased LTL attrition was observed in those born before 37 weeks of GA, as well as in those who gained weight as adults in a cohort study in India. GA was positively associated with offspring RTL, although there was no significant association of offspring RTL with body size at birth including birthweight, birth length, and birth BMI. Conditional BMI gain at 2 and 11 years was not associated with RTL. BMI gain at 29 years was negatively associated with RTL. Born SGA was not associated with RTL in adulthood [107].

## 4. Discussion

In the current study, the systematic review of traditional methodology was rejected since the combination of the search keywords “Maternal nutrition”, “Preterm birth” and “Placental and/or newborn telomeres”, returned zero results. For this reason, the keywords A: maternal nutrition, B: PTB, and C: placental and/or newborn TL, were searched in dyads as follows: is concept “A” related to concept “B”, “C” to “A”, and “C” to “B”; thus, leading to the assumptive inference of A being related to B and C. It is impossible, however, to find the chronological sequence of the exposure and the outcome, but evidence of associations between these factors would imply either a causal or a correlational relationship [72].

The data from the relevant literature search led to the final selection of 40 observational studies. The overall results indicate a relationship between maternal nutrition and PTB, as well as maternal nutrition and newborn and placental TL. However, the evidence for the relationship between PTB and TL, was inconclusive. The schematic representation of the aggregated key findings can be seen in Figure 2.

### 4.1. Maternal Nutrition and PTB

In terms of maternal nutrition and PTB, all 21 studies included in this review indicated that there are nutrients (either through dietary intake or supplementation) and eating patterns associated with increased risk of PTB, while others play a beneficial and/or protective role against PTB.

Nutritional status, as assessed by selected biomarkers, is associated with higher risk of PTB and GA at birth. In particular, protein level in maternal blood plasma is inversely associated with a risk of PTB and positively associated with gestational duration in the third trimester of pregnancy, particularly in the female fetus [92]. Dietary protein is a macronutrient well known for its relationship with fetal health in the past decades [108,109,110,111,112,113]. However, few studies have investigated the role of protein in PTB risk, and the results are inconclusive, especially when examining protein supplementation and PTB incidence [114]. By using selected valid and reproducible biomarkers of maternal protein plasma levels additionally to dietary intake assessments—which present numerous challenges to obtaining accurate nutritional status—significant confounders are omitted (i.e., quantification of intake, self-report bias, pathological causes of reduced absorption, etc.) [92].

Another nutritional marker examined for its role in PTB is folate. Higher maternal serum folate concentration at approximately the start of the third trimester, was associated with a longer duration of gestation and lower risk of PTB, while little or no correlation was seen between serum B6 and B12 and PTB [89]. Supportive of the above findings regarding folate, one more study showed that higher plasma levels of the nutrient are associated with lower risk of PTB [84]. The relationship between folate levels and PTB was further supported in a recent meta-analysis of folic acid and the risk of PTB. The meta-analysis evidence indicates that high early pre-conception maternal folate levels are significantly associated with a lower risk of PTB. Moreover, protective action against spontaneous PTB was seen following daily use of 400 μg folate periconceptionally [115]. However, folate supplementation does not appear to protect against iatrogenic PTB or prolonged spontaneous rupture of fetal membranes [81]. The large cohort study by Wu et al. [82] showed similar results with folic acid periconceptional supplementation being associated with a lower risk of PTB. Specifically, the earlier women started taking their folic acid supplements prior to pregnancy (i.e., at least 3 months before their last menstrual period), the more likely this was to reduce the risk of PTB, compared with women who started taking folate later (i.e., 1–2 months before their last menstrual period) [82]. An earlier systematic review suggested that supplementation is associated with a significant reduction in the risk of PTB, but only when being initiated immediately after conception [23]. Although further RCTs are warranted to establish the exact relationship between folate and PTB, its beneficial effect on the reduction of birth defects, supports the notion that it is necessary in adequate amounts during the periconceptional period.

Vitamin D is another micronutrient studied for its effects on PTB risk in three studies included in this review. Firstly, the cross-sectional study by Christoph et al. [90] showed that the lower the vitamin D maternal level the higher the GA at birth. A second cross-sectional study in Indonesia, comparing a PTB group with a term birth group, found that the latter had a higher placental concentration of 25(OH)D [86]. In addition, in a third nested case–control study in Bangladesh, vitamin D deficiency in pregnant women was associated with an increased risk of PTB [91]. An earlier meta-analysis had already shown evidence of the association between maternal circulating 25-OHD deficiency (rather than insufficiency) and an increased risk of PTB. To enhance the argument, the meta-analysis concludes that vitamin D supplementation alone during pregnancy may reduce PTB risk [116]. However, RCTs to date have unanimously failed to ascertain the prophylactic effect of vitamin D against PTB, with the emergence of conflicting evidence [117]. It is suggested that low levels of vitamin D may reflect poor general maternal health status. Thus, priority should be given to the attainment of general health, rather than vitamin D supplementation per se [118].

Maternal iron deficiency anemia was positively associated with PTB [88]. In particular, hemoglobin (Hb) levels of less than 11 mg/dL were directly associated with PTB, low birth weight, fetal mortality, and a low Apgar score in the 1st and 5th minutes of birth [88]. In a recent systematic review and meta-analysis, iron deficiency anemia was indeed found to be a contributing factor towards PTB during the first trimester, but not in the second and third trimester [119]. When accounting for iron during pregnancy, it is imperative to consider its critical role for embryonic development and fetal growth when transported through the placenta from the mother to the fetus. Furthermore, since iron cannot be synthesized by the body, sufficient iron absorption from dietary sources is very important for both mother and fetus [120]. Still, little is known about the iron states in the mother, the placenta, and the fetus, and which mechanisms responsible for iron transport contribute towards PTBs. A recent study attempting to characterize maternal and fetal iron metabolism in pregnant women with PTB found a dysregulated iron homeostasis in both sides and a disordered placental iron equilibrium, which were presumed to account for the compromised fetal iron supply [121]. Prevention or treatment with either intravenous iron supplementation or oral medication showed no significant differences in maternal and neonatal outcomes, thus emphasizing the need for nutritional correction of iron-deficient states during pregnancy [122].

The effect of the availability of other micronutrients on PTB risk is far more complex than the previously discussed folate, vitamin D, and iron. In the Irwinda et al. [86] cross-sectional study, lower concentrations of all-trans retinoic acid (AtRA) in maternal serum were inversely associated with PTB, while PTB was also positively associated with higher serum concentrations of heavy metals such as mercury and lead. Also, compared with the PTB group, the term birth group had higher maternal serum concentrations and higher placental concentrations of manganese, iron, copper, zinc, selenium, AtRA, lower placental concentrations of mercury and lead, and in cord blood, higher concentrations of copper and AtRA [86]. A current scoping review also documented a higher incidence of PTB with lead and cadmium exposures. The findings for mercury and arsenic exposures were, however, inconclusive. The most common pathways through which heavy metals and metalloids lead to increased risk of PTB are placental oxidative stress, epigenetic modifications, inflammation, and endocrine disruptions [123].

In hair samples, nutritional trace metal(loid) concentrations, especially Fe and Zn, are negatively associated with PTB and, surprisingly, the endocrine disrupting metal(loid) Hg was also negatively associated with PTB [85]. Conversely, PTB was associated with higher maternal serum concentrations of copper and zinc in a nested case–control study in Malawi, indicating perhaps the difference between types of sampling source of the biomarker (e.g., blood serum and plasma, cellular fractions, tissue, etc.) [87]. Due to the conflicting results, and the low-to-moderate certainty evidence, it is suggested that zinc supplementation may reduce PTB risk in women with low zinc intake, low levels, or poor nutrition [114].

Any type of multivitamin supplementation during the third trimester of pregnancy (but not earlier) was associated with a significant reduction in PTB among non-Hispanic and African women in a large USA cohort study [83]. In agreement is the case–control study by Olapeju et al. [84], where multivitamin supplement intake of at least three times/week throughout pregnancy was significantly associated with a reduction in the risk of PTB. In both studies described above, no significant association was recorded between preconceptional multivitamin intake and risk of PTB. Conversely, in a systematic review and meta-analysis, the use of multivitamins and adverse birth outcomes in high-income countries, did not change the risk of PTB [124]. However, the authors stressed the need for additional data on multivitamin intake in pregnancy, from randomized controlled trials or large cohort studies, controlling for multiple confounders.

In terms of dietary patterns, religious fasting (such as during the Ramadan period of the Islamic religion) in the second trimester of pregnancy, increased the risk of very PTB [75]. This cohort had a large population of 78,109 births, making the evidence robust. In contrast, the recent review and meta-analysis that examined the effect of Ramadan fasting during pregnancy on perinatal outcomes of 5600 births, showed no effect on PTB [125]. Studies indeed have shown conflicting results on the effect of fasting on PTB, perhaps due to differences in the timing of exposure [125]. More well-designed observational studies with large samples investigating all types of fasting during all trimesters of pregnancy are required to shed light on its impact on maternal and fetal health [126].

A dietary pattern during pregnancy based on fruits, vegetables, and rice protected women against PTB and SGA infants, compared with dietary patterns based on seafood and noodles or based on processed meat, cheese, and pasta. Nevertheless, the fruit/vegetables/rice pattern contributed towards the development of LGA infants [79]. Supportive of this evidence is the study by Teixeira et al. [78], in which the group of women giving birth prematurely, had a below the recommendations for pregnant women by the Portuguese Directorate General of Health consumption of certain foods (i.e., dairy products, cereals, vegetables, and fruits). In another study examining the effect of dietary patterns on PTB severity levels, a pattern rich in rice and nuts lowered the risk of very/moderate PTB compared with late PTB or term birth, while a high dietary consumption of starchy foods was associated with the most severe level of PTB incidence [127]. Dairy products, on the other hand, seem to have a controversial effect. It is argued that an increased consumption of dairy products has a protective effect against PTB, when combined with a maternal normal BMI [76]. Otherwise, the increased consumption of dairy products, especially milk, as well as cereals, eggs, and Cantonese soups, and the ‘Fruits, nuts, and Cantonese desserts’ groups, compared with vegetable consumption, had a strong correlation with PTB [77].

On the same note, in the prospective cohort of Martin et al. [80], maternal diet quality was associated with PTB, where a greater adherence to a healthy dietary pattern, such as the DASH diet, reduced the risk of PTB, especially during the second trimester. Conversely, a greater adherence to a dietary pattern of poorer quality, such as preprocessed or fast foods, high-saturated fat foods, and confectionary, increased the risk of PTB. Comparing five different food patterns, namely “Obesogenic”, “Traditional”, “Intermediate”, “Vegetarian”, and “Protein”, Miele et al. [93] performed a nested case–control study in Brazil and reported that a diet rich in protein increases the probability of developing preeclampsia and PTB. However, the anthropometric classification of obesity had a greater impact on preeclampsia (PE) and gestational diabetes mellitus (GDM), in contrast to PTB and SGA outcomes, suggesting that the effect of the dietary pattern on PTB is dependent upon other anthropometric characteristics. In a recent review, which included 40 observational studies, the dietary patterns during pregnancy associated with a lower risk of PTB, were also characterized by high consumption of vegetables, fruits, as well as whole grains, fish, and dairy products [11].

In general, the evidence indicates that a Western-type diet, high in meat and fats and low in fruits and vegetables, is associated with an increased risk of induced PTB [128]. In contrast, adherence to a Mediterranean and/or healthy diet pattern during pregnancy appears to be associated with a reduced risk of PTB [129,130]. It may also be that the protective effect of increasing the intake of foods associated with a Mediterranean or healthy dietary pattern is more important than totally excluding highly processed food, fast food, confectionary, and snacks [131].

Similar results were noted with nutritional scoring systems. In the cohort of Parisi et al. [73], a positive association between a healthy maternal nutritional score and GA at birth was reported, while a significant increase in the risk of PTB was associated with low nutritional scores. However, the single-food-items score calculations were not associated with either early placental markers or complex pregnancy outcomes. Also, a high-Norwegian Fit for Delivery diet score, either pre-pregnancy or in early pregnancy, was inversely associated with gestational weight gain (GWG) and PTB risk [74]. Examining dietary patterns as a whole, independent of the manner of assessment, has emerged as a holistic approach for capturing the complex interactions between nutrients and foods [132]. In summary, the evidence shows that maternal nutrition during pregnancy, as assessed through dietary patterns, is a major determinant for birth outcomes and, consequently, offspring health outcomes in later life [133].

Surprisingly, no recent studies were found that exclusively examine maternal BMI and the risk of PTB. Previous evidence showed that either a high or low BMI is associated with increased risk of PTB. However, when limited to developing countries, low BMI was not significantly associated with PTB [118]. BMI is an obscure measure of nutritional status and health, especially since it does not take into account body composition (e.g., muscle mass, bone density, etc.) and racial and sex differences [134], and thus is not considered as a valid factor accounting for PTB risk. In contrast, recent findings from the ongoing prospective “Mamma & Bambino” study (Catania, Italy), suggest that gestational gain weight, rather than BMI per se, affects maternal TL. Specifically, women with adequate gestational gain weight showed longer TL than those who gained inadequate weight. Additionally, the TL of cfDNA exhibited a U-shaped relationship with weight gain during pregnancy, suggesting that increased weight gain also heralds negative effects [135]. Mechanisms by which inadequate or excessive weight gain affect TL are yet to be elucidated but could be related to chronic inflammation and oxidative stress in utero.

The above evidence indicates that dietary patterns linked to enhanced anti-inflammatory and antioxidant properties, containing a variety of nutrient dense, unprocessed or minimally processed foods, that reduce the risk of developing nutritional deficiencies, are the most appropriate for ascertaining a healthy period of pregnancy and fetal development, as well as the avoidance of adverse birth outcomes.

### 4.2. Maternal Nutrition and Placental and Newborn Telomeres

The findings from exploring the association between maternal nutrition and placental and infant TL conclude that diet has an important role in whether or not placental TL will be maintained. Specifically, the level of maternal plasma vitamin D seems to be positively correlated with placental TL, thus contributing to TL maintenance or reduction of TL attrition. Conversely, an inverse effect was observed for BMI, body fat percentage, vitamin B12 and placental TL, but no associations were apparent for cord blood TL [97].

Further studies support the protective role of maternal plasma vitamin D in maintaining the newborn’s TL. In two studies included in this review, the concentration of serum vitamin D during pregnancy and maternal energy intake were positively correlated with the neonate’s LTL at birth [47,98]. The pleiotropic effects of vitamin D in the organism, and especially its immunomodulatory effects, may be one mechanism by which it is protective against telomere attrition [136]. More recently, this evidence was strengthened by a cohort in Hong Kong where 25(OH)D, D3, and D3 epimer, both in utero and at birth, impacted childhood LTLs [137]. Furthermore, insufficient maternal vitamin D (25(OH)D) has been associated with increased offspring risk for many diseases and later life adverse outcomes [138].

In contrast, Herlin et al. found no association between antioxidant maternal serum levels of zinc, selenium, folate, and vitamin D3, and maternal or newborn TL [101]. The contradicting results of maternal vitamin D levels and their effect on offspring TL may be due to the differences in the timing of vitamin D level assessments during pregnancy. For example, Daneels et al. showed the importance of maternal nutrition early in pregnancy, and in particular the first trimester, being associated with TL at birth [98]. In support of the above, another study in Switzerland showed that the effects of vitamin D are more pronounced during the earlier gestational period [139].

Three studies included in this review investigated the effect of folate on fetal and placental TL [46,99,100]. A positive association was observed between maternal folic acid levels early in pregnancy and newborn cord blood TL [46]. This evidence suggests that fetal telomeres exhibit developmental plasticity and show that maternal nutrition can affect or even “program” this system. The Louis-Jacques et al. study [100] also showed a positive association between umbilical cord red blood cell folate levels and fetal TL at birth, while a possible association between maternal folic acid supplementation during pregnancy and longer newborn TL was suggested in the cohort study by Fan et al. [99]. The role of folate in DNA methylation and oxidative stress are proposed mechanisms through which it influences TL regulation in the offspring [140]. In contrast, in the study by Kim et al., dietary intake of dietary folate equivalents, assessed by 24 h recalls, was not associated with fetal TL [47]. A recent systematic review of the effect of maternal diet and offspring TL also concluded that higher circulating maternal folate and 25-hydroxyvitamin D3 concentrations were associated with longer offspring TL, adding to the equation a protective effect for higher maternal dietary caffeine intakes [141]. To date, the data regarding higher dietary intake of folate, in regard to offspring TL regulation remain contradictory [103].

Another study examining maternal micronutrient status and TL in maternal serum, cord blood, amniotic fluid, and placenta, showed that only magnesium deficiency is negatively associated with maternal RTL. Furthermore, a positive association between maternal intake of magnesium and TL of cfDNA from amniotic fluid was seen, while results on other micronutrients (i.e., vitamin B1 and iron) were marginally significant [142]. Moreover, Nsereko et al. concluded that lower ferritin, soluble transferrin receptor levels, and retinol-binding protein levels are associated with longer maternal TL [143]. The effect of various micronutrients (both intakes and status) and TL in PTB remains obscure and demands further investigation.

The effects of dietary fat and its biomarkers have also been investigated. Shortened TL among fetuses exposed to maternal high fat consumption during pregnancy, after accounting for the effects of potential covariates, was recorded in the Salihu et al. study [96]. However, according to the Liu et al. study, concentrations of total n3- PUFAs and docosahexaenoic acid (DHA) in maternal erythrocytes, are closely correlated to infant TL and the telomerase reverse transcriptase (TERT) promoter methylation [94]. Contradictory evidence for maternal PUFA status and its relation to offspring TL was recorded in the cohort study by Yeates et al. [95]. There were no clear associations recorded for either prenatal or postnatal PUFA status with offspring TL. However, the higher prenatal n6:n3 PUFA ratio was associated with longer TL in mothers. Maintaining higher levels of maternal n3 PUFAs during pregnancy may help to conserve the offspring TL, accompanied by potential benefits in offspring long-term health.

In terms of maternal dietary patterns and offspring TL regulation later in life, the recent—first of its kind—systematic review examining the impact of dietary intake factors on TL in childhood and adolescence, suggests that a higher consumption of fish, nuts and seeds, fruits and vegetables, leafy and cruciferous vegetables, olives, legumes, polyunsaturated fatty acids, and an antioxidant-rich diet might positively affect TL. Conversely, a high intake of dairy products, simple sugars, sugar-sweetened beverages, cereals, especially white bread, and a diet high in glycemic load were associated with enhanced TL shortening in the offspring during childhood and adulthood [144].

Offspring sex also appears to affect associations between maternal nutritional intake and/or status and TL. Indeed, TL in female newborns was shown to be more susceptible to variation from maternal TL and vitamin B12 levels, while in newborn male TL from parental age, maternal education, plasma fasting glucose, DGLA%, and IGFBP3 levels [65]. Thus, demographic factors such as offspring sex and maternal ethnicity, seem to also affect the relationship between maternal nutrition and offspring TL, and therefore should be treated as confounding factors in all relevant investigations [103].

It is relatively well established in both animal and human studies that nutrition plays a profound role on DNA integrity, epigenetic mechanisms, and TL regulation [32,145,146,147]. Indeed, various nutrients influence TL through mechanisms depicting their role in cellular functions including inflammation, oxidative stress, DNA integrity, and methylation, as well as telomerase activity [145]. However, evidence from human studies examining the impact of maternal nutrition on newborn TL remains scarce, not allowing definite conclusions to be reached. Since TL at birth represents an individual’s initial setting of TL and predicts later life TL [148], the effects of maternal diet on offspring TL should be thoroughly investigated. Longitudinal studies assessing nutritional intake and/or status—with various indices including nutrient biomarkers—and exploring possible associations with offspring and placental TL regulation, are warranted. Upon ascertaining the potential nutritional determinants of offspring TL regulation, TL status and attrition rates could become valuable as markers of future chronic disease risk.

### 4.3. PTB and Placental and Newborn Telomeres

As for the association of PTB with both placental and the newborn TL, only six studies were found. TL appears to be highly variable in newborn infants. In particular, PTB infants were found to have longer TL than full-term infants, while TL was significantly negatively correlated with GA and birth weight. A positive correlation was also observed between maternal age and the telomere/beta-globin single-copy gene (T/S) ratio, indicating that the older the mother, the longer the newborn’s telomeres, possibly indicating a compensatory mechanism. In the same study, longitudinal assessment of preterm infants who had TL measurements available at the age of 5 years, suggests that TL attrition rate is negatively correlated with increasing GA [61]. The findings of the systematic review and meta-analysis by Niu et al. [62] are in agreement. In this study, IUGR was associated with shorter placental TL, while PTB with longer birth TL, but only as measured by qPCR and not by the method of telomere restriction fragment (TRF) analysis. This may explain the lack of correlation of TL measurements when assessed by different laboratory methods, which may account for controversies in study results. The TRF method has been proposed as a method of high accuracy, as it results in lower variation than the PCR method [149]. However, possibly due to its time consuming, labor intensive, costly, and expertise-demanding characteristics, TRF is used in very few studies compared with PCR.

Furthermore, in the Niu et al. [62] review and meta-analysis, IUGR was associated with shorter birth TL only when birth TL was measured in the placenta, but not in newborn blood. TL is sensitive to the types of tissues used for measurement, as previous studies have reported the placental TL as being relatively longer than cord blood TL [150]. Nevertheless, irrespective of the studied biological matrix (i.e., cord leukocytes or placenta), newborn TL measurements remain predictive of TL in leukocytes at the age of 4 years [43]. Although, the choice of sample type is a complex matter, involving numerous issues such as attainability, availability and ethics, it must be made clear that when comparing studies using different biological samples, this could lead to discordant conclusions.

In the study by Colatto et al., membranes from term labors, also showed TL reduction compared with those from the PTB group. Telomerase activity did not change in fetal membranes irrespective of pregnancy outcome. The authors suggested that telomere shortening in fetal membranes is indicative of senescence associated with the triggering of labor at term [105]. The longer TL recorded in PTB offspring is a surprising finding since PTB has been associated with early aging phenotypes [151], but may be explained by the fewer cellular replications and reduced DNA turnover in the final few weeks of gestation, otherwise missed when born preterm [152]. Indeed, in a twin study, placental TL gradually decreased with GA, indicating that enhanced TL attrition is more prominent with advancing GA [153]. Adjustments for GA should be made when measuring birth TL in all studies between groups of different maternal nutritional exposures.

On the other hand, the study by Saroyo et al. showed that telomere T/S ratio of the placenta did not differ between PTB and term labor despite difference in GA, suggesting perhaps similar TLs in the timing of sampling between the two groups, due to early telomere shortening in PTB that mimics the term placenta [104]. The timing of sampling is indeed of extreme importance when assessed in PTB infants, especially if one considers the effects of trauma in the NICU, and the previously reported enhanced TL shortening of preterm infants experiencing these adversities following birth [68].

In contrast, GA has also been positively associated with offspring RTL. Longitudinal data indicate an increased LTL attrition in those born before 37 weeks of GA, as well as in those who gained weight as adults (29 years) but conditional BMI gain at 2 and 11 years, was not associated with RTL [107]. Previous studies have shown, however, that birth TL differs according to body size and the GA of the newborn [154,155]. Maternal anthropometric characteristics may perhaps explain, to an extent, why studies reach inconclusive outcomes when comparing TLs from spontaneous PTB offspring, compared with term-born infants [106].

Another longitudinal cohort followed premature newborns into their adulthood by studying the TL in their saliva and assessing their lung function. A positive correlation between TL and abnormal lung airflow in an adult population born prematurely was observed, but there was no apparent association with perinatal causes. It is speculated that there is continuous oxidative stress of the airways, which leads to persistent inflammation, lung function alteration, and increased sensitivity for chronic obstructive pulmonary disease [70]. Furthermore, TL attrition rates are associated with diseases characterized by increased inflammation and oxidative stress [156], thus affecting TL regulation in the offspring in an independent manner.

Evidence to date has confirmed that newborn TL strongly predicts child and adulthood TL [43,149], while it is the strongest predictor of TL change over time [157,158]. Thus, TL measured at any point in life is jointly determined by TL at birth and subsequent TL attrition in later life, with a rapid attrition rate before the age of 5 years old and then remains relatively stable for the remainder of the life course [159]. TL at birth indeed has been reported to vary as much as 3000–5000 bp inter-individually, a proportion that is well above the overall TL attrition occurring in the whole lifespan [160]. Therefore, birth TL is extremely important for the lifetime health trajectory and aging pace rhythm determines the interindividual variation of adult TL, which in turn has been proposed as a biomarker and potential contributor to the development of aging-related chronic diseases [161,162].

### 4.4. Maternal Nutrition, Placental Newborn Telomeres, and PTB

From the above evidence, it appears that maternal nutrition affects PTB risk, partly through its influence on maternal TL (Figure 3. Representation of the associations). Indeed, the telomere-regulated-clock mechanism determines the length of gestation, leading to the onset of labor (parturition), and at the same time PTB is a major determinant of offspring TL regulation. The strength of the associations and the extent of the influence from covariates remains to be elucidated in future research. Furthermore, the question of whether maternal TL is simply a biomarker of maternal nutritional status, and in effect PTB risk, or a causative factor of PTB through nutritional habits and status, remains unanswered to date. Even so, studies have supported the idea that TL may be used as a novel biomarker, especially in combination with traditional indicators for the prediction of PTB and, thereafter, preterm health, such as weight development in preterm neonates [163]. Reliable biomarkers with high prognostic value are highly sought-after for efficient decision-making in clinical settings.

Following telomere expansion at the beginning of pregnancy, TL in the placenta and the fetal membranes, gradually shortens throughout the remainder of gestation. The rate of telomere shortening can be affected by nutritional practices, either directly, for example through their inflammatory/anti-inflammatory and oxidant/antioxidant properties [41,164], or indirectly by influencing other aspects of health also associated with TL, such as mental health (MH). Indeed, nutrition is associated with MH outcomes, such as a higher risk for developing depression and anxiety [165,166] while, on the other hand, MH may affect nutritional status (e.g., through appetite regulation) [167]. Additionally, evidence shows a link between various types of MH disorders and TL, including stress and anxiety [168,169,170], bipolar disorder [171,172], and psychotic disorders [173,174]. Perhaps therefore, MH through its effect on maternal dietary habits, independently affects maternal and offspring TL. To our knowledge, no studies to date however, have examined the bi-directional relationship between maternal nutrition and MH, assessing simultaneously TL regulation in the premature offspring.

As previously mentioned, due to the critical role of birth TL in determining later life TL [155,175,176,177], it is intriguing to explore the potential mediation role of TL biology, underlying the relationship between intrauterine exposures and aging-related diseases by primarily comparing TL at birth between PTB and term birth outcomes. The current findings thus emphasize the necessity for prospective, longitudinal studies that investigate the relationship between placental/newborn TL and maternal nutrition—particularly during conception and pregnancy—in PTB risk. With the use of machine learning, algorithms, and big data analysis, insights of the exact relationship of the aforementioned factors may be provided, also considering personal characteristics, genetics profiles, MH, way of life, and general wellbeing.

### 4.5. Limitations of the Study

This study has several limitations. First, studies included in this review used different research methodologies and assessment methods, variable set outcomes and exposures, and geographical, demographic and anthropometric population characteristics. Population sizes are furthermore extremely variable between studies.. Secondly, many studies did not adjust for sex and race/ethnicity. Thirdly, exposures, especially those with key roles in the outcomes of studies, for example MH status (i.e., subclinical depression, traumatic experiences, eating disorders, etc.), remained unexplored or unadjusted for. Fourthly, the well-documented difficulty leading to compromised validity and reliability of nutritional intake and status assessments, was not adequately addressed in the majority of the studies.

Also, we searched for relevant publications only in two electronic databases and did not include any unpublished studies or articles, such as meeting abstracts and dissertations, which might have introduced publication biases. Finally, we included solely articles written in English, which might also have caused the omission of articles published in other languages.

Future studies using individual participant data with adjustments for sex, race/ethnicity, anthropometry, and other potential confounders, are imperative for valid comparisons. In addition, cross-sectional studies do not provide information of causality, because of the unclear temporality between the exposure and outcome and this should be considered in review and meta-analysis studies.

## 5. Conclusions

There is a relationship between maternal nutrition, placental-newborn TL, and PTB. Specifically, maternal nutrition influences PTB risk both positively and negatively, to an extent through its influence on maternal TL. Furthermore, maternal TL independently affects PTB risk, while at the same time PTB appears as a major determinant of offspring TL regulation. The strength of the associations and the extent of the influence from various covariates remains largely unexplored. The question also remains of whether maternal TL is simply a biomarker of maternal nutritional status and in effect PTB risk, or a causative factor of PTB, through nutritional practices and status.

From a nutritional point of view, a diet high in carbohydrates and low in protein in the third trimester and high in fat in the first trimester of pregnancy, was associated with PTB and SGA infants. Micronutrients and their adequate intake may also play an important role in PTB risk. In particular, iron, zinc, B vitamins (folic acid, B6, B12) seem to affect PTB prevention and folic acid, vitamin D, B12, and n3 PUFAs are factors that aid in the maintenance of both the placental and infant TL. However, the strength of the evidence is not adequate to reach definite conclusions and the formulation of clinical guidelines.

In addition, placental telomeres play an important role in PTB and act as a biomarker of both the mother’s and the newborn’s nutritional status. Indeed, placental and maternal TL is reported as being shorter in PTBs compared to full-term pregnancies, thus TL may be considered to be a biomarker of prognostic value for the adverse outcome of premature labor. Maternal leukocyte TL at the beginning of pregnancy, on the other hand, constitute a prognostic indicator of maternal biological aging and, further, indirectly, a prognostic indicator of the offspring’s health from fetal life to the early post-natal period. Since premature infants exhibit a more rapid TL attrition in early life, while a measurable decline in TL during NICU hospitalization has been reported, the rate of TL change in early life may convey information about prematurity and NICU exposures that carry both immediate and long-term health risks.

The findings of this review signify the urgent need for further research that will assess the relationships between the abovementioned factors simultaneously, especially in longitudinal observational studies, following populations from pregnancy to the offspring’s’ adulthood. Preventative medicine and future research may use this information regarding TL (maternal and placental) as well as, maternal nutritional status, for the screening of high-risk preterm delivery or pregnancies, quantification of premature offspring adversity burden in early life, and recording of their long-term health consequences in offspring adulthood.

## Figures and Tables

**Figure 1 nutrients-15-04975-f001:**
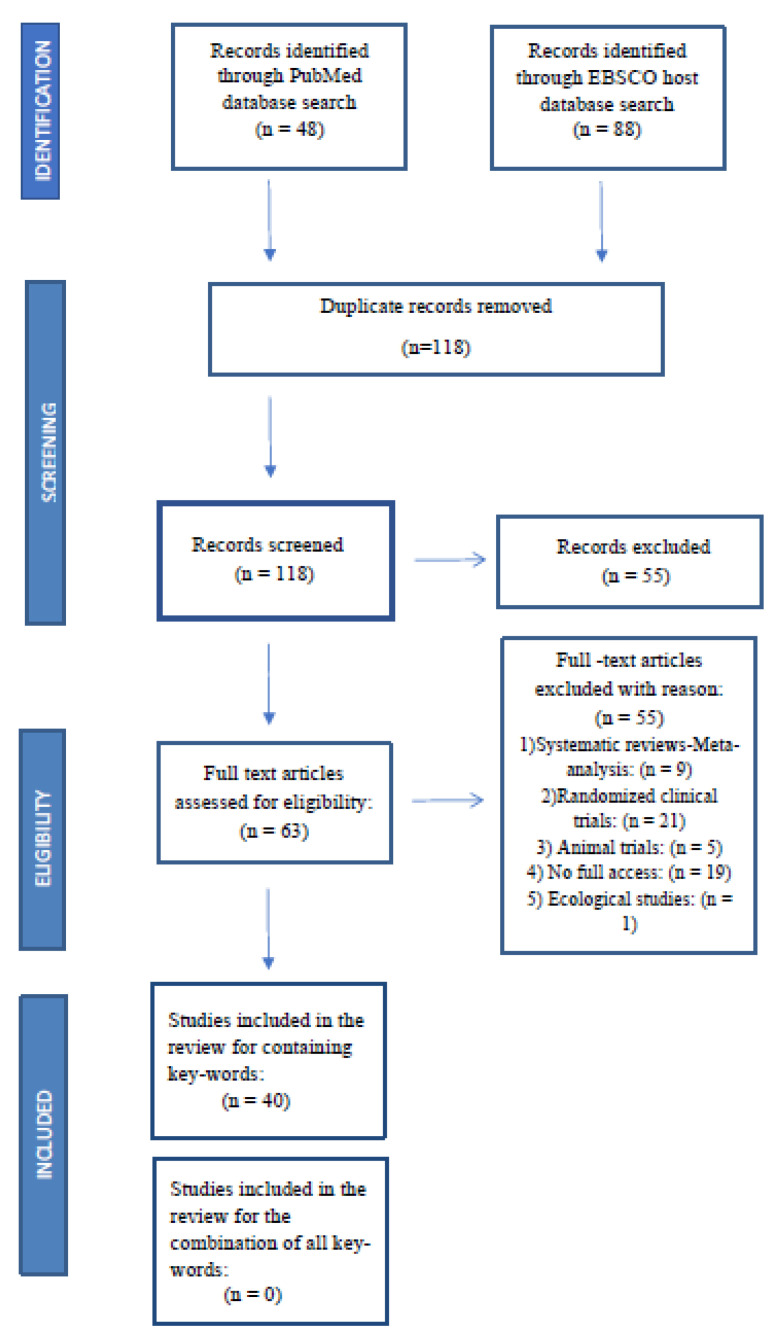
PRISMA flow diagram 2009 [71]. Methodology used for study selection.

**Figure 2 nutrients-15-04975-f002:**
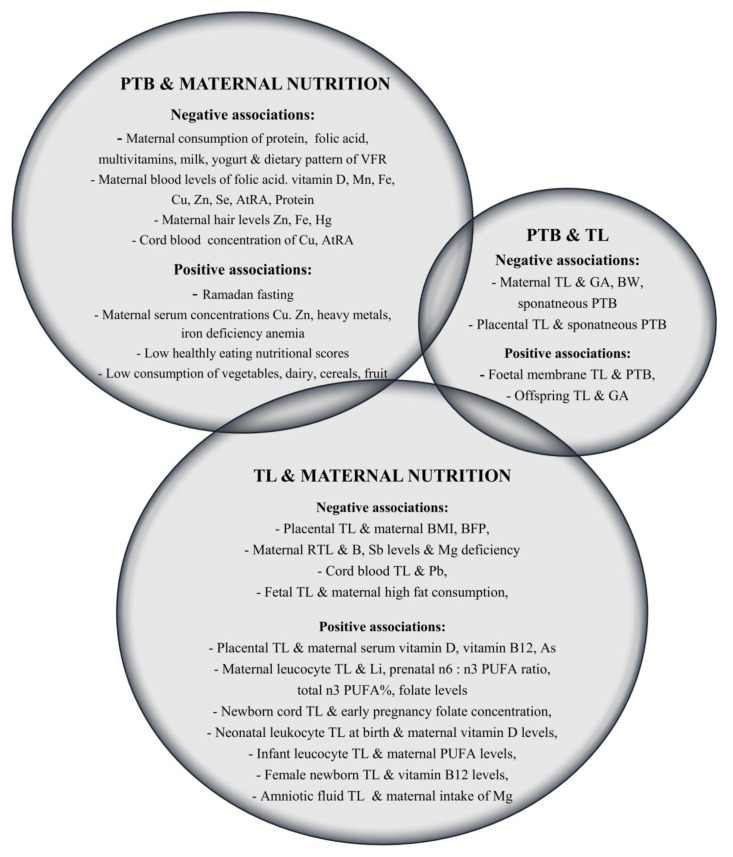
Schematic representation of aggregated key findings regarding the associations between PTB, maternal nutrition, and placenta or fetal TL. PTB: preterm birth, TL: telomere length, RTL: relative TL, VFR: vegetables/fruits/rice, Zn: zinc, Fe: iron, Hg: mercury, AtRA: all-trans retinoic acid, Mn: manganese, Pb: lead, As: arsenic, Cu: copper, Se: selenium, Sb: antimony, Li: lithium, PUFAs: polyunsaturated fatty acids, Mg: magnesium, GA: gestational age, BW: birth weight, BFP: body fat percentage.

**Figure 3 nutrients-15-04975-f003:**
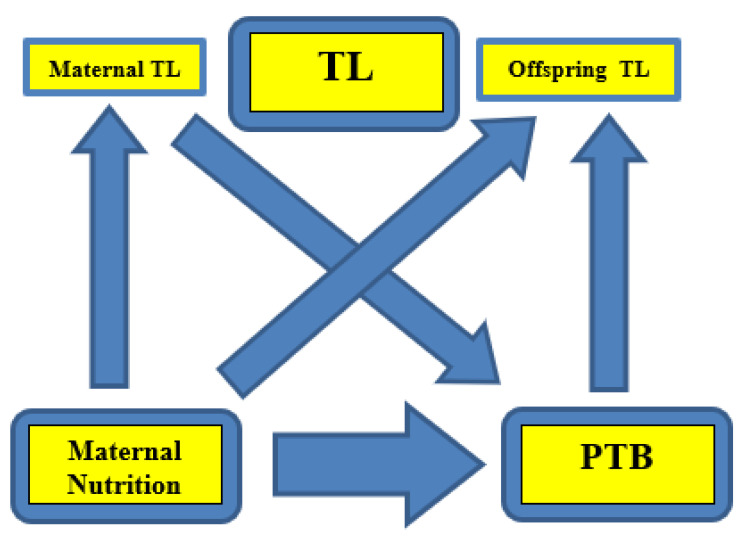
Representation of the associations between PTB, maternal nutrition and TL. Maternal nutrition affects PTB risk, partly through its influence on maternal TL (and in effect placenta TL). On the other hand, maternal TL independently affects PTB risk and, at the same time, PTB is a major determinant of offspring TL regulation. PTB: preterm birth, TL: telomere length.

**Table 1 nutrients-15-04975-t001:** Studies associating maternal nutrition and preterm birth.

Research	Type of Study	Country	Population/ Subject Characteristics	Outcome
1. Parisi et al., 2020 [73]	Cohort	Italy	112 pregnant women (11 + −13 + 6 weeks of gestation); Absence of pathology; Singleton pregnancy; Natural conception; Healthy neonates.	Positive association between maternal nutritional score and GA at birth; Significant increase in the risk of PTB associated with low nutritional scores; The single food items of the score calculation were not associated with both early placental markers and complex pregnancy outcomes.
2. Hillesund et al., 2018 [74]	Cohort	Norway	591 nulliparous pregnant women 18 years or older; 20 weeks pregnant or less; Singleton pregnancy; Absence of pathology, disabilities, and substance abuse.	A higher diet score either pre-pregnancy or in early pregnancy was protectively associated with excessive GWG and PTB risk; The protective association with high birthweight was confined to pre-pregnancy diet and with preeclampsia to early pregnancy diet; No association between pre-pregnancy diet score and preeclampsia.
3. Tith et al., 2019 [75]	Cross-sectional	Canada	78.019 births of Arabic speaking women; Three trimesters of pregnancy during the Ramadan period; Absence of pathology.	Ramadan fasting during the second trimester of pregnancy was associated with increased risk of very PTB (28–31 weeks gestation).
4. Kriss et al., 2018 [76]	Cohort	Mexico	965 pregnant women; 18–22 weeks gestation—follow up 2 years post-birth 18–39 years old; Absence of pathology.	No overall association between prenatal yogurt consumption and PTB; In non-overweight women, higher prenatal yogurt consumption was associated with reduced PTB risk.
5. Lu et al., 2018 [77]	Cohort	China	7352 pregnant women <20 weeks gestation; Singleton pregnancy; Absence of pathology.	Women in the high consumption of ‘Milk’ group had greater risk of PTB, spontaneous PTB, and late PTB, than those in the ‘Vegetables’ group; Compared with women in the ‘Vegetables’ group, those in the ‘Cereals, eggs, and Cantonese soups’ and ‘Fruits, nuts, and Cantonese desserts’ groups had an increased risk of late PTB; Maternal pregnancy diet with frequent consumption of milk and less frequent consumption of vegetables is found to be associated with increased risk of PTB among Chinese women.
6. Teixeira et al., 2023 [78]	Cross-sectional	Portugal	60 Portuguese women with a diagnosis of high-risk pregnancy; Birth before 33 weeks GA.	In very PTB, pregnancy-induced hypertension was associated with increased consumption of pastry products, fast food, bread, pasta, rice, and potatoes; Only bread consumption had a weak but statistically significant association with pregnancy-induced hypertension in a multivariate analysis.
7. Chia et al., 2016 [79]	Cohort	Singapore	923 pregnant women; Absence of pathology.	The VFR pattern is associated with a lower incidence of PTB; The VFR pattern was also associated with a higher birth weight, higher ponderal index, and increased risk of LGA deliveries.
8. Martin et al., 2015 [80]	Cohort	USA	3143 pregnancies at 26–29 week of gestation	Greater adherence to a healthy dietary pattern, such as the DASH diet, reduced the risk of PTB; Greater adherence to a dietary pattern of poorer quality, such as factors 2 (i.e., high intakes of beans, corn, French fries, hamburgers or cheeseburgers, white potatoes, fried chicken, spaghetti dishes, cheese dishes, cornbread or hushpuppies, processed meats, biscuits, and ice cream) and factors 3 (i.e., high consumption of collard greens, coleslaw or cabbage, red processed meats, fried chicken, fried fish, cornbread or hushpuppies, eggs or egg biscuits, gravy, whole milk, and vitamin C-rich drinks), increased the risk of PTB.
9. Li et al., 2014 [81]	Cohort	China	207 936 singleton live births; Delivered at GAs of 20–42 weeks; Absence of pathology.	Daily consumption of 400 μg folic acid alone during the periconceptional period is related to a reduced risk of spontaneous PTB. These reduced risks were greater for early age spontaneous PTBs.
10. Wu et al., 2021 [82]	Cohort	China	201,477 pregnant women aged 18–49; Singleton livebirths.	Periconceptional supplementation with folic acid was associated with a lower risk of PTB; Women who started taking folic acid at least 3 months before their last menstrual period were more likely to reduce the risk of PTB.
11. Johnston et al., 2016 [83]	Cohort	USA	62.443 pregnant women ≥18 weeks of gestation; Singleton pregnancy; Absence of pathology.	Any multivitamin use in the last 3 months of pregnancy is associated with a significant reduction in PTB among non-Hispanic and African women.
12. Olapeju et al., 2019 [84]	Case—control	USA	7675 mother–infant dyads; Cases: mother–infant dyads, singleton, live, low-birth weight (<2500 g) or preterm infants (<37 weeks of gestation) regardless of birth weight; Controls: mother–infant dyads with singleton, live, term infants with birth weight of ≥2500 g or more.	Multivitamin supplement intake of at least three times/week throughout pregnancy was significantly associated with a reduction in the risk of PTB; Use during the third trimester was associated with a greater reduction in PTB risk than use in the first trimester; No significant association between pre-conceptional supplement intake and PTB; Higher plasma folate levels were associated with lower risk of PTB.
13. Ren et al., 2022 [85]	Case-control study	China	509 pregnant women: Controls: 415 pregnant women with birth at term and Cases: 82 pregnant women with PTB; Singleton pregnancy; Absence of pathology.	Negative correlation between levels of NTMs, and PTB; Fe and Zn especially may protect against PTB occurrence; Potential protective effect of Hg.
14. Irwinda et al., 2019 [86]	Cross sectional	Indonesia	51 pregnant women undergoing birth (term group: 25, PTB group: 26); Singleton pregnancy; Absence of pathology.	PTB associated with lower concentrations of micronutrients in maternal serum; PTB associated with a higher concentration of heavy metals such as Hg and Pb; Compared with the PTB group, the term birth group had higher maternal serum concentration of the AtRA serum; Compared with the PTB group, the term birth group had higher placental concentrations of Mn, Fe, Cu, Zn, Se, AtRA, 25(OH)D, and lower placental concentrations of Hg and Pb; Compared with the PTB group, the term birth group had higher concentrations of Cu and AtRA in cord blood.
15. Chiudzu et al., 2020 [87]	Nested case–control	Malawi	181 pregnant women: 90/181 (49.7%) term and 91/181 (50.3%) PTBs; Singleton pregnancy; Presenting in spontaneous labor with intact membranes; Absence of pathology.	Cu maternal serum concentrations were above the upper normal limit in both term and PTB groups; PTB was associated with higher maternal serum concentrations of Cu and Zn.
16. Perveen & Soomro, 2016 [88]	Cohort	Pakistan	234 pregnant women aged 20–35 and at 35–42 weeks of gestation; Singleton pregnancy; Absence of pathology.	Fe deficiency anemia positively associated with PTB, low newborn BW, fetal mortality, and low Apgar score in the 1st and 5th minutes of birth.
17. Chen et al., 2015 [89]	Cohort	Singapore	999 pregnant and <14 weeks gestation & post-partum women, aged 18–50 years old; Only parents of the same ethnicity; Absence of pathology.	Higher maternal folate concentrations at approximately the start of the third trimester significantly associated with longer duration of gestation and lower risk of PTB; Little or no correlation between high folic acid levels and SGA; Little or no correlation between B6 and B12 vitamin levels and PTB or SGA.
18. Christoph et al., 2020 [90]	Cross-sectional	Switzerland	1382 pregnant women; Supplemented with 600 IU/d orally throughout the pregnancy, and at least 1000 IU/d in cases of deficiency.	No association between the 25(OH)D serum level and PTB, preeclampsia, postdate pregnancy, miscarriage, intrauterine growth restriction, bacterial vaginosis, mode of delivery, or neonatal BW and length; The lower the maternal vitamin D level, the higher the GA at birth.
19. Tahsin et al., 2023 [91]	Nested case–control	Bangladesh	930 pregnant women at 8–19 weeks of gestation; 262 PTB and 668 term births.	Vitamin D deficiency is common in Bangladeshi pregnant women and is associated with an increased risk of PTB.
20. Xiong et al., 2021 [92]	Cohort	China	3382 mother–newborn second trimester pairs and 3478 mother–newborn third-trimester pairs; Absence of pathology.	Third trimester MTP level, inversely associated with PTB risk and positively associated with gestational duration; The effects of the 3rd trimester MTP level on PTB risk and gestational duration, were stronger in women carrying female offspring than those carrying males.
21. Miele et al., 2021 [93]	Nested case–control	Brazil	1165 nulliparous pregnant women between 19 and 21 weeks of gestation; Singleton pregnancies; Absence of pathology; Absence of medication or supplementation.	The likelihood of adverse outcomes was higher in non-white (*p* < 0.05) obese women with high protein consumption; The anthropometric classification of obesity had a greater impact on PE and GDM, in contrast to PTB and SGA.

[n]: Reference number. PTB: preterm birth, SPTB: spontaneous preterm birth, BW: birth weight, SGA: small for gestational age, LGA: large for gestational age, KTR: kynurenine to tryptophan ratio, PE: preeclampsia, MTP: maternal plasma total protein, VFR: vegetables/fruits/rice, SfN: seafood and noodles, PCP: processed meat, cheese, pasta, EDMs: endocrine-disrupting metal(loid)s, Mn: manganese, Cu: copper, Se: selenium, Pb: lead, Hg: mercury, Se: selenium, Pb: lead, NTMs: nutritional trace metal(loid)s, Zn: zinc, Fe: iron (Fe), Cu: copper, GWG: gestational weight gain, UAC: upper arm circumference, GDM: gestational diabetes mellitus, 25(OH)D3: 25-hydroxyvitamin D_3_.

**Table 2 nutrients-15-04975-t002:** Studies associating maternal nutrition with placental and newborn telomeres.

Research	Type of Study	Country	Population/ Subject Characteristic	Outcome
1.Entringer et al., 2015 [46]	Cohort	USA	119 mother–newborn dyads enrolled at 9.5 weeks gestation; Absence of pathology or fetal abnormality; Use of folic acid supplements.	Maternal total folate concentration in early pregnancy was significantly and positively associated with newborn cord blood TL.
2. Kim et al., 2018 [47]	Cross-sectional	Seoul	106 mother–newborn dyads; Pregnant women in the third trimester; Absence of maternal and fetal pathology.	Positive correlation between the concentration of vitamin D of the pregnant woman and the neonatal leukocyte TL (β = 0.33, *p* < 0.01).
3. Chen et al., 2022 [65]	Cohort	Singapore	950 mother–offspring dyads; In their first trimester of pregnancy; Participants with homogeneous parental ethnic backgrounds.	TL in female newborns more susceptible to variation from maternal TL, mental health, and vitamin B12 levels; Variation in newborn male TL was explained more by their parental age, maternal education, plasma fasting glucose, DGLA%, and IGFBP3 levels. Overall, maternal TL strongly associated with antenatal factors, especially metabolic health and nutrient status.
4. Liu et al., 2022 [94]	Cohort	China	274 mother–newborn dyads; Pregnant women enrolled during the third trimester; Absence of pathology.	Positive association of the concentrations of DHA and total n-3 PUFAs in maternal erythrocytes with DNA methylation of the TERT promoter in the cord blood instead of the placenta; Maternal PUFAs closely correlated to infant TL and TERT promoter methylation, differently affected by maternal n3 PUFAs between the cord blood and the placenta; High levels of maternal n3 PUFAs during pregnancy maintain offspring TL.
5. Yeates et al., 2017 [95]	Cohort	Seychelles	229 mothers enrolled in a 28-week gestation and followed through delivery; Their children (at 5 years of age).	No clear associations of both prenatal or postnatal PUFA status and methylmercury exposure with child TL: Higher prenatal n6:n3 PUFA ratio was associated with longer TL in mothers.
6. Salihu et al., 2018 [96]	Cohort	USA	62 women upon admission for delivery; Singleton pregnancy; No indication of congenital malformations.	Shortened TL among fetuses exposed to maternal high fat consumption during pregnancy, after accounting for the effects of potential covariates.
7. Vahter et al., 2020 [97]	Cohort	Argentina	99 Pregnant women in 1st trimester; Singleton pregnancy.	Maternal BMI, BFP, and vitamin B-12 were inversely associated; 25(OH)D3 was positively associated with placental TL; No association between the above factors was observed with cord blood TL.
8. Daneels et al., 2021 [98]	Cohort	Belgium	108 pregnant women in the first trimester or women who are trying to conceive; Absence of pathology; Singleton pregnancy.	Positive association of maternal vitamin D concentration (diet and/or supplement) in the first trimester with TL of the newborn at birth.
9. Fan et al., 2022 [99]	Cohort	China	746 mother–newborn pairs < 16 weeks of pregnancy; Singleton gestation.	Possible association between maternal folic acid supplementation during pregnancy with longer newborn TL was suggested.
10. Louis-Jacques et al., 2016 [100]	Cohort	USA	96 maternal–fetal dyads; Singleton gestation; No indication of congenital malformations.	Positive association between umbilical cord RBC folate and fetal TL at birth.
11. Herlin et al., 2019 [101]	Cohort	Argentina	169 pregnant women in 1st trimester; Singleton pregnancy.	More associations with relative TL in maternal blood leukocytes during pregnancy (inverse associations with boron and antimony, positive association with lithium), than in the placenta (positive with arsenic) and cord blood (inverse with lead); Nutritional antioxidants (Zn, Se, folate, Vit D3) did not affect the associations.
12. Magnano San Lio et al., 2022 [102]	Cohort	Italy	174 pregnant women, in the 4th–20th gestational week; Singleton pregnancy; Absence of pathology and pregnancy complications; Planned follow-up of their children at delivery and up to 2 years of age.	Mg deficiency negatively associated with maternal RTL after adjusting for the same covariates; Positive association between maternal intake of Mg and TL of cfDNA from amniotic fluid, while results on other micronutrients (i.e., vitamin B1 and Fe) were marginally significant.
13. Myers et al., 2021 [103]	Cohort	USA	96 maternal–fetal dyads; Singleton gestation; No indication of congenital malformations.	Positive association between maternal vitamin C intake and fetal TL.

[n]: reference number. STI: sexually transmitted infection, sTfR: soluble transferrin receptor, RBP: retinol binding protein, PUFA: polyunsaturated fatty acids, cfDNA: cell-free DNA, IGFBP3: insulin-like growth factor binding protein 3, CRP:C-reactive protein, Fe: iron, Zn: zinc, Se: selenium, DGLA: dihomo-γ-linolenic acid, AGP: ambulatory glucose profile. PTB: preterm birth, BMI: body mass index, BFM: body fat mass, BFP: body fat percentage, RTL: relative telomere length, TL: telomere length, GA: gestational age, RBC: red blood cell, TERT: telomerase reverse transcriptase, Mg: magnesium, DHA: docosahexaenoic acid, TERT: telomerase reverse transcriptase, DGLA: dihomo-gamma-linolenic acid, 25(OH)D3: 25-hydroxyvitamin D_3_.

**Table 3 nutrients-15-04975-t003:** Studies associating preterm birth with placental and newborn telomeres.

Research	Type of Study	Country	Population/ Subject Characteristics	Outcome
1. Vasu et al., 2017 [61]	Case-control	United Kingdom	78 newborns (47 premature <32 gestational weeks and 31 term newborns); Absence of antenatal or postnatal severe congenital malformation or unlikely to survive.	RTL significantly negatively correlated with GA and BW in preterm infants; RTL highly variable in newborn infants; Preterm infants at term equivalent age, have significantly longer TLs than term born infants; Positive correlation between maternal age and T/S ratio; Longitudinal assessment in preterm infants with TL measurements available at birth and term age (n = 5) suggests that, TL attrition rate is negatively correlated with increasing GA; RTL was significantly shortest in the term born control group compared with both PTB groups and longest in the PTB at birth group; In addition, TL was not significantly different between preterm infants sampled at birth and those sampled at term equivalent age.
2. Hadchouel et al., 2015 [70]	Cohort	France	274 adults; Group 1: 236 adults born prematurely; Group 2: 38 adults born full-term.	No apparent association with perinatal events; Positive association between TL and abnormal lung airflow in the population born prematurely.
3. Saroyo et al., 2021 [104]	Cross—sectional	Indonesia	67 placentas (34 placentas from preterm and 33 placentas from term birth were included); Singleton pregnancy; Absence of pathology and pregnancy complications.	Similar TL due to early telomere shortening in PTB that mimics the term placenta; 8-OHdG and HMGB1 do not correlate with placental telomere ratio; HMGB1 from the placenta of both PTB and term labor has no significant difference; Equal telomere T/S ratio of the placenta from PTB and term labor.
4. Colatto et al., 2020 [105]	Cross sectional	Brazil	80 singleton pregnancies; Fetal membrane samples collected from (1) pregnant women with pPROM, (2) PTB with intact membranes, (3) term labor and (4) TNL; Absence of pathology and pregnancy complications.	Fetal membranes from the term group showed TL reduction, compared with those from the other groups; Telomerase activity did not change in fetal membranes irrespective of pregnancy outcome; Telomere shortening in fetal membranes is suggestive of senescence associated with triggering of labor at term.
5. Farladansky-Gershnabel et al., 2023 [106]	Cross sectional	Israel	19 women with spontaneous term pregnancies and 11 with SPTB; Absence of preterm premature rupture of membranes or induced preterm labor; Absence of pregnancy pathology or pregnancy complications; Nulliparous; Absence of fetal malformations.	Maternal blood and placental samples from SPTB had shorter telomeres and increased Gal-3 expression compared with the spontaneous term pregnancies group.
6. Tarik et al., 2019 [107]	Cohort	India	1309 offspring; Mean maternal age 39.08 y (±3.29); BMI gain at 2, 11, and 29 years.	GA positively associated with offspring RTL; No significant association of offspring RTL with body size at birth including BW, birth length, and birth BMI; Conditional BMI gain at 2 and 11 years not associated with RTL; BMI gain at 29 year was negatively associated with RTL; Born SGA was not associated with RTL in adulthood; Increased LTL attrition was observed in those born before 37 weeks of GA, as well as in those who gained weight as adults.

[n]: reference number. LTL: leucocyte telomere length, RTL: relative telomere length, pPROM: preterm premature rupture of membranes, TL: telomere length, GA: gestational age, PTB: preterm birth, BMI: body mass index, T/S ratio: telomere/single copy gene ratio, OR: odds ratio, GA: gestational age, SGA: small for gestational age, BW: birth weight, EVT: extravillous trophoblast, SPTB: spontaneous preterm birth, Gal-3: galanine, TNL: term not in labor, BMI: body mass index, HMGB1: high mobility group box 1, 8-OHdG: 8-hydroxy-2’-deoxyguanosine.

## Data Availability

No new data were created or analyzed in this study.

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
