# Peer review of "Preterm Birth and Its Association with Maternal Diet, and Placental and Neonatal Telomere Length"

_nutrients, 2023, doi:10.3390/nu15234975_

Round 1
Reviewer 1 Report
Comments and Suggestions for Authors
The article titled "Preterm birth and its correlation with maternal diet, placental factors, and neonatal telomere length," authored by Lisa et al., provides a comprehensive overview of the relationship between preterm birth, maternal nutritional intake, and telomere length in both the placenta and infants. The authors have meticulously presented and analyzed current research spanning the years 2013 to 2023. In general, this manuscript demonstrates a systematic and well-structured approach that is likely to capture the interest of Nutrients' readership. Nevertheless, there are a few issues that require attention before considering the manuscript for publication.
Firstly, the readability of the figures needs improvement. Secondly, there is redundancy in the content, with certain information being repeated in both the Results and Discussion sections. To enhance clarity, it is recommended to place tables and figures in their respective sections. For instance, Table 1 could be appropriately positioned under Section 3.2.
Author Response
Dear Reviewer 1, thank you kindly for your important and detailed reviewing contributions towards our review article. Please find our answers to your comments bellow:
- Firstly, the readability of the figures needs improvement.
Answer: Figure 2 and Figure 3 have been altered to increase readability. In figure 2, the grouping of results are presented in a homogenous format, while in figure 3, the arrows clearly indicate associations between maternal and offspring TL.
- Secondly, there is redundancy in the content, with certain information being repeated in both the Results and Discussion sections.
Answer: The following content has been removed from the discussion section:
- Page 22, lines 601-604
- Page 22, lines 620-622
- Page 23, lines 669-670
- Page 23, lines 680-682
- Page 24, lines 690-695
- Page 24, lines 704-708
- Page 24, lines 721-722
- Page 24-25, lines 738 – 740
- Page 25, lines 747-751
- To enhance clarity, it is recommended to place tables and figures in their respective sections. For instance, Table 1 could be appropriately positioned under Section 3.2.
Answer: Tables and figures placed in their corresponding sections in the text.

Reviewer 2 Report
Comments and Suggestions for Authors
This review article describes that the association between PTB, maternal nutrition and pla-15 cental-infant TL. Maternal TL independently affects PTB risk, and at the same time PTB is a major determinant 20 of offspring TL regulation.
Inadditon, recent research on the relevance of telomere length and gestational weight gain or gestational age should be further discussed:
1. The Relationship between Telomere Length and Gestational Weight Gain: Findings from the Mamma & Bambino Cohort. Biomedicines. 2022 Jan; 10(1): 67.
"Women with adequate GWG showed longer telomere length than those who gained weight inadequately. An early effect of GWG on telomere length of cfDNA, which could represent a molecular mechanism underpinning the effects of maternal behaviours on foetal well-being."
2. Cord blood telomere shortening associates with increased gestational age and birth weight in preterm neonates. Exp Ther Med. 2021 Apr;21(4):344.
"telomere length may be a novel biomarker alone or in combination with traditional indicators for the prediction of weight development in preterm neonates."
Author Response
Dear Reviewer 2, thank you kindly for your important and detailed reviewing contributions towards our review article. Please find our answers to your comments bellow:
In addition, recent research on the relevance of telomere length and gestational weight gain or gestational age should be further discussed:
- The Relationship between Telomere Length and Gestational Weight Gain: Findings from the Mamma & Bambino Cohort. Biomedicines. 2022 Jan; 10(1): 67. "Women with adequate GWG showed longer telomere length than those who gained weight inadequately. An early effect of GWG on telomere length of cfDNA, which could represent a molecular mechanism underpinning the effects of maternal behaviours on foetal well-being."
Answer: Added in page 23, lines 651-656, reference number: [135]
- Cord blood telomere shortening associates with increased gestational age and birth weight in preterm neonates. Exp Ther Med. 2021 Apr;21(4):344. "telomere length may be a novel biomarker alone or in combination with traditional indicators for the prediction of weight development in preterm neonates."
Answer: Added in page 27, lines 838-842, reference number: [163]
